# Network Analyses of Integrated Differentially Expressed Genes in Papillary Thyroid Carcinoma to Identify Characteristic Genes

**DOI:** 10.3390/genes10010045

**Published:** 2019-01-14

**Authors:** Junliang Shang, Qian Ding, Shasha Yuan, Jin-Xing Liu, Feng Li, Honghai Zhang

**Affiliations:** 1School of Statistics, Qufu Normal University, Qufu 273165, China; shangjunliang110@163.com; 2School of Information Science and Engineering, Qufu Normal University, Rizhao 276800, China; dingqian19@126.com (Q.D.); jiayouyss@126.com (S.Y.); sdcavell@126.com (J.-X.L.); 3School of Computer Science and Technology, Xidian University, Xi’an 710071, China; lifeng_10_28@163.com; 4College of Life Science, Qufu Normal University, Qufu 273165, China

**Keywords:** differentially expressed genes (DEGs), interaction network, papillary thyroid cancer (PTC), Normalized Centrality Measure (NCM)

## Abstract

Papillary thyroid carcinoma (PTC) is the most common type of thyroid cancer. Identifying characteristic genes of PTC are of great importance to reveal its potential genetic mechanisms. In this paper, we proposed a framework, as well as a measure named Normalized Centrality Measure (NCM), to identify characteristic genes of PTC. The framework consisted of four steps. First, both up-regulated genes and down-regulated genes, collectively called differentially expressed genes (DEGs), were screened and integrated together from four datasets, that is, GSE3467, GSE3678, GSE33630, and GSE58545; second, an interaction network of DEGs was constructed, where each node represented a gene and each edge represented an interaction between linking nodes; third, both traditional measures and the NCM measure were used to analyze the topological properties of each node in the network. Compared with traditional measures, more genes related to PTC were identified by the NCM measure; fourth, by mining the high-density subgraphs of this network and performing Gene Ontology (GO) and Kyoto Encyclopedia of Genes and Genomes (KEGG) enrichment analysis, several meaningful results were captured, most of which were demonstrated to be associated with PTC. The experimental results proved that this network framework and the NCM measure are useful for identifying more characteristic genes of PTC.

## 1. Introduction

The morbidity rate of papillary thyroid carcinoma (PTC) ranks the highest among thyroid cancer. Exploring the cause and underlying molecular events of PTC has become a hot topic [1], and various methods for analyzing characteristic genes of PTC have emerged. There are many methods for exploring the characteristic genes of PTC. Kim et al. [2] proposed to select differentially expressed genes (DEGs) by using paired *t*-test, linear models for microarray data, and significance analysis of microarrays. Chevillard et al. [3] used microarrays containing probes to hybridize with RNA, which came from normal and tumor thyroid samples labeled with Cy3-dUTP and Cy5-dUTP, respectively. Jarzab et al. [4] proposed a potent molecular classifier able to discriminate between PTC and nonmalignant thyroid, which used a support vector machines-based technique. However, most of the existing methods focus on analyzing differences between genes in two experimental conditions or using statistical methods to analyze interactions between functionally similar genes. It becomes increasingly difficult to identify the characteristic genes from analysis results with the diversity and high dimensionality of biosequencing data, which makes the advantages of network mining methods more obvious.

The methods of network mining can systematically reflect the interactions between biomolecules; thus, it is beneficial for researchers to fully understand how various biomolecules in biological cells interacting and performing biological function processes. He et al. [5] proposed a method to identify molecular markers related to the clinicopathological features of renal cell carcinoma through a weighted gene co-expression network, but multiple thresholds need to be set through the hub gene of the Weighted Gene Co-expression Network Analysis (WGCNA) calculation module. Zhang et al. [6] proposed a method for analyzing gene interaction enrichment, which combines the knowledge of gene pathways to identify interaction effects on enriched genes. This method ignores genes that have no direct effect on expression, but interact with other genes to affect expression. Kamran et al. [7] proposed a new measure of the topology properties of the network; however, this measure ignores the relationship between the various factors of the nodes in the formula and the importance of the nodes. Wei et al. [8] proposed the Length-Net-Driver (LNDriver) method to construct a gene-gene interaction network, which integrates mutation data and expression data into the network, and finally uses the greedy algorithm to select driver genes. The genes obtained by this method only considering the local features and do not consider the role in the global network. In addition, existing network methods to mine characteristic genes are mostly limited to considering only local features or global features.

In this paper, we propose a framework to identify the characteristic genes of PTC. First, DEGs identified from multiple datasets were integrated and used to build an interaction network. The nodes in the network represent genes, the edges between two nodes indicating that these two nodes interact with each other. The weight of an edge was quantified by a comprehensive scoring mechanism, which was formed by comprehensive data, data mined from PubMed abstract text, database data, and results predicted using bioinformatics methods. Next, in order to identify the characteristic genes, a new measure Normalized Centrality Measure (NCM) is proposed in this paper, which takes into account both local and global features of nodes. Compared with the existing measures, characteristic genes identified by NCM were more biologically significant. Moreover, we used the previously proposed Hypergraph Construction and High-Density Subgraph Detection (HC-HDSD) method and performed Gene Ontology (GO) term and Kyoto Encyclopedia of Genes and Genomes (KEGG) pathway enrichment analysis to supplement the experimental results. Our research framework and experimental results provide reliable molecular markers for early detection and prognosis.

## 2. Methods

### 2.1. Datasets

Four profile datasets, i.e., GSE3467 [9], GSE3678 [10], GSE33630 [11], and GSE58545 [12], were downloaded from the GEO database (http://www.ncbi.nlm.nih.gov/geo/) for this study. Platforms, sample numbers, and citations of these four datasets are shown in Table 1.

### 2.2. Identification of Differentially Expressed Genes

Raw data were converted into an expression matrix and the *t*-test method in limma package [13] was subsequently used to identify DEGs between PTC tissue samples and normal thyroids [14]. In the expression matrix, one row represents one sample and one column represents one gene. A gene which satisfied the conditions of *p*-value (adjusted by Benjamini & Hochberg) < 0.05 and logFC > 2 (FC, fold change) was identified as an up-regulated gene. A gene which satisfied conditions of *p*-value (adjusted by Benjamini & Hochberg) < 0.05 and logFC < −2 was identified as a down-regulated gene. All identified up-regulated genes and down-regulated genes were integrated into DEGs. Description of *p*-value and logFC can be seen in [13].

### 2.3. Construction of Interaction Network of Differentially Expressed Genes

The identified DGEs were utilized to construct a gene-gene interaction network. The nodes represented genes, the tag names of nodes correspond to gene names, and the edges between nodes indicate interactions between the two nodes. Sources of network edges contain experimental data, data mined from PubMed abstract text, database data, and results predicted using gene contiguous, gene fusion, phylogenetic profile, and gene co-expression based on chip data and other bioinformatics methods, which were based on the STRING database (http://string-db.org) [15]. The composite score, which was systematically calculated by different methods described above, were used as weights of edges in the constructed network. Isolated nodes that were not connected to other nodes were deleted. In addition, the size of a node indicated the number of neighbors the node had, that is, the larger the node, the more edges that were connected to this node.

### 2.4. Analyses of Differentially Expressed Genes Interaction Network

#### 2.4.1. Traditional Topological Properties

There were many measures to quantify the topology properties, currently. The following traditional five measures, which we selected, are most widely used to quantify the topological properties of a network. The definitions of them are as follows.

The Degree Centrality (DC) of node v represents the number of nodes which are directly connected to v in the network. It is defined as:(1)$$DC(v)=\sum_{j\in G}\frac{a_{vj}}{n-1}$$
where n is the total nodes in the network and $a_{vj}$ represents that node v and node j are directly connected. DC is the most basic topological property to reflect local features of the node in the network.

Betweenness Centrality (BC) is a measure to quantify the number of times the node v appearing in the shortest path between other nodes, which is defined as:(2)$$BC(v)=\sum_{i\neq j\neq v\in V}\frac{\sigma_{ivj}}{\sigma_{ij}}$$
where $\sigma_{i,j}$ represents the number of the shortest paths from node i to node j, $\sigma_{ivj}$ represents the number of paths through node v. BC reflects the global features of the node. The higher the value of BC, the more important the node is in keeping the network tightly connected.

Closeness Centrality (CC) is a measure to quantify the average shortest distance from one node to all other nodes in the network. The CC of node v is defined as follows:(3)$$CC(v)=\frac{1}{n-1}\sum_{j\neq v\in V}d_{vj}$$
where n is the number of nodes, $d_{vj}$ represents the shortest distance from node v to node j. The CC measure reflects the extent of node v closing to the “center” of the network. The smaller the value of CC, the closer the node is to the center, which represents the global features of the node.

Clustering Coefficient (CLC) embodies the dense connection nature between some nodes, which is defined as:(4)$$CLC(v)=\frac{2n}{k(k-1)}$$
where k represents the number of neighbors of node v, and n represents the number of sides connected between k neighbors of node v. A higher value of CLC indicates that the node is more closely connected to its neighbors.

The Eccentricity Centrality (EC) is the maximum of the distance from one node to others in the network. The eccentricity of node v in the network G is defined as:(5)$$EC_{G}(v)=\frac{1}{\max_{V\in G}\{(dist(v,u))\}}$$
where u is a node in the network G. dist(v,u) is the largest geodesic distance between the node v and other nodes. The smaller the value of EC, the more important the node is in the network.

#### 2.4.2. Normalized Centrality Measure

Considering that almost all the existing measures consider only one of the local features and the global features, the results obtained may be relatively one-sided. Therefore, in this paper, we propose a new measure, Normalized Centrality Measure (NCM), by combining the three traditional measures: degree centrality, betweenness centrality, and closeness centrality to identify important genes by calculating the value of NCM for node v:(6)$$NCM(v)=(\frac{BC(v)}{BC_{\max}}+\frac{DC(v)}{DC_{\max}})/\frac{CC(v)}{CC_{\max}}$$
where DC(v) is the degree centrality of node v and $DC_{\max}$ is the maximum value of degree centrality in the network CC(v) is the closeness centrality of node v and $CC_{\max}$ is the maximum value of closeness centrality in the network; BC(v) is the betweenness centrality of node v and $BC_{\max}$ is the maximum value of betweenness centrality in the network. NCM comprehensively reflects both the global and local features of nodes. The larger the value of the NCM(v) of the node v, the more important the node is. From a theoretical analysis, this measure is meaningful.

#### 2.4.3. Detection of High-Density Subgraphs

In addition to calculating the topological properties of each node, this paper also introduces the HC-HDSD method to extract high-density subgraphs, which was proposed in our previous work. The high-density subgraph was actually the interactions between several genes that may cause PTC, which is in the form of cluster.

First, the Maximal Clique Centrality (MCC) method [16] to detect the hub nodes and their subgraphs:(7)$$MCC(v)=\sum_{C\in S(v)}(|C|-1)!$$
where S(v) is the set contains the maximal clique group of v, C is a subset of S(v), (|C|−1)! is the product of all positive integers, which were less than the number of nodes in the maximal clique C.

Then, based on the subgraphs obtained by the above method, the NCMine method [17] is used to further detect meaningful high-density subgraphs. The threshold cliqueness used to determine whether to extract the high-density subgraph is defined as follows:(8)$$cliqueness(G)=\frac{E(G)}{C_{V(G)}^{2}}$$
where G is the new module after adding a node, E(G) is the number of edges of the new module, and V(G) is the number of nodes of the new module.

### 2.5. Gene Ontology and Pathway Enrichment Analysis

Gene ontology analysis is used to annotate genes as well as gene products, and identify biological characteristic for DEGs. GO annotations are performed on the DEGs by using a DAVID online tool (https://david.ncifcrf.gov/) [18], which was an essential foundation for the success of high-throughput gene function analysis. In addition, KEGG pathway analysis of DEGs is performed using the KOBAS online analysis database (version 3.0) (http://kobas.cbi.pku.edu.cn/) [19]. In this paper, we analyzed the DEGs, which were derived from the integrated up-regulated and down-regulated genes from four different microarray datasets; *p*-value < 0.05 was set as the cut-off criterion.

## 3. Results and Discussions

### 3.1. Identification of Differentially Expressed Genes

There are 77, 53, 123, 238 DEGs obtained by the limma package from the above four datasets, among which there were 30, 28, 90, and 127 up-regulated genes, as well as 47, 25, 43, and 111 down-regulated genes, respectively. The distribution of DEGs is shown in the following volcano map. The red circles represent up-regulated genes, the green circles represent down-regulated genes, and the gray circles represent non-DEGs. Figure 1a–d show DEGs identified from datasets GSE3467, GSE3678, GSE33630, GSE58545, respectively.

### 3.2. Analyses of Differentially Expressed Genes Interaction Network

A complex interaction network is obtained from the STRING database to describe interactions between genes, with a total of 501 DEGs including 275 up-regulated genes and 226 down-regulated genes. The nodes in the network represent genes. After removing the duplicate values and the individual nodes that do not connect to other nodes, there were a total of 261 nodes in this network. Otherwise, edges between two nodes represent that these two nodes are interacted; the tags on nodes represent the gene names. The size of the node represents the number of neighbors directly connected to the node. The larger the node, the more neighbors connected to this node. A gene-gene interaction network is established for all identified DEGs, which is shown in the Figure 2.

### 3.3. Analyses of Characteristic Genes

The topological properties DC, CC, BC, CLC, EC, and NCM of each node in the above network were calculated. The larger the values of DC, BC
CLC and NCM, the more important the node was; the smaller the values of CC and EC, the more important the role that this node played in the network. The values of DC, BC, CLC and NCM were sorted from large to small; the values of CC and EC were sorted from small to large. The relationship between the top 40 values of these topological properties and their corresponding genes is shown in Figure 3. Figure 3a–e show the values of traditional topological property we calculated and their corresponding genes. Figure 3f shows the top 40 genes calculated using our newly proposed measure NCM; among them, there are 34 genes that have been confirmed in the existing literature to be related to the production of PTC. The accuracy was 85%. The values of the logFC and *p*-value of these 40 genes are shown in the tables in the attached file, and 31, 30, 20, 18, and 22 genes were found by calculating the traditional measure of topological properties DC, BC, CC, CLC, and EC. The accuracies were 77.5%, 75%, 50%, 45%, and 55%, respectively. Table 2 shows genes that have been confirmed to be involved in the production of PTC. These genes belong to the top 40 gene calculated by using DC, BC, CC, CLC, EC, and NCM measures. In Table 2, genes with bold font have been recorded in GeneCard database (http://www.genecards.org/), of which was confirmed that they are associated with PTC. In addition, other genes have been confirmed to be related to PTC by referring to the existing literature.

Analyses of top 40 genes calculating by the newly proposed topological property NCM measure were as follows. Qiu et al. [20] suggested that *FN1*, *ICAM1* and *TIMP1* play an important roles in the progression of PTC; *COL1A1* [21], *COL5A1* [21], and *PLAU* [21] were confirmed to provide therapeutic biomarkers of PTC. *MET* [22], and *SERPINA1* [22] were screened as common DEGs, which may be related to PTC. *LCN2* [23] transcript levels in tumor tissues were significantly higher in PTC compared with normal tissues. *SDC4* [2] and *LRRK2* [2] were validated that they have a significant genetic difference with normal tissue. *ECM1* [24] and *KIT* [24] were identified as DEGs that contain valuable candidate genes to differentiate related tumor types on the molecular level. Hyun Sook et al. [25] confirmed that the expression of *SLC34A2* and *COMP* increase in PTC compared with normal tissue. *CXCL12* and *IRS1* were identified as central genes in [26], they may be key genes for PTC. *CFD* [27] may be helpful for the development of gene panel for PTC diagnosis. *PCSK2* [28] could be used to identify the benign and malignant follicular thyroid neoplasia. *MUC1* [29] was demonstrated to be an important oncogene in PTC and make an important role in therapeutic cancer vaccine development. *ALDHIA1* [30] was down-regulated in PTC tissues. *SLC26A4* [30] and other genes revealed follicular variant of PTC *CD36* [31] was reported to correlate with neovascularization of PTC DCN [32] could help for thyroid tumors to better define a tumor signature and be useful for thyroid tumor treatment. *LRP2* [33] makes effects for thyroid through expressing retained intracellularly. *AGTR1* [34] methylation signatures which involved in the logistic regression model, may related to thyroid cancer. *APOE* [35] was identified as hub gene which may be the predictive risk marker of PTC in the process of progression and prognosis. *NMU* and *LPAR5* may be the targets for thyroid cancer treatment, which are abnormally expressed in thyroid cancer tissues [36]. *ADH1B* [37] could be the target autoantigens in thyroid cancer and play potential clinical role. Kai-Chun et al. [37] demonstrated that *GNA14* involves in the regulation of the thyroid function and autoimmunity. *ALDH1A3* [37] may be the factor of mediating cancer stemness in thyroid cancer. *EGR1* [38] may have a marked effect on thyroid cancer. Stein [39] identified that *ANK2* gene shows a high expression level in PTC.

### 3.4. Analyses of High-Density Subgraphs

At present, most methods for identifying DEGs are focused on single gene. We propose a new idea to find the combination of genes, that is, extracting the high-density subgraphs. The high-density subgraphs are closely connected internally, and the connections with other nodes are sparse. High-density subgraphs extracted from the constructed network are shown as Figure 4. The first high-density subgraph is composed of *MUC1* [29], *LGALS3* [22], *ICAM1* [20], *TIMP1* [20], *CXCL12* [26], *MET* [22], and *FN1* [20], which are shown in Figure 4a. Consistent with what we described before, all these genes have been reported to be related to PTC. We hypothesize that *MUC1*, *LGALS3*, *ICAM1*, *TIMP1*, *CXCL12*, *MET*, and *FN1* may contribute to PTC in the form of a high-density subgraph that interact with each other. The second high-density subgraph is composed of *GABBR2* [24], *CXCR7* [52], *CCL21* [49], *CXCL12* [26], *LAPR5* [36], *NMU* [36], *GNA14* [37], *AGTR1* [34], *EDN3*, and *AVPR1A*, which are shown as Figure 4b. Among them, *GABBR2*, *CXCR7*, *CCL21*, *CXCL12*, *LAPR5*, *NMU*, *GNA14*, and *AGTR1* have been reported that they are related to PTC, and other two genes *EDN3* and *AVPR1A* have not been reported to directly lead to PTC. Therefore, the interaction of these genes in the form of a high-density subgraph may lead to PTC. The third high-density subgraph is composed of *ICAM1* [20], *EGR1* [38], *KIT* [24], *NCAM1*, *CXCL12* [26], *PROM1*, *PGF*, and *MET* [22], which are shown as Figure 4c. Among them, *ICAM1*, *EGR1*, *KIT*, *CXCL12*, and *MET* have been reported that they are related to PTC; however, *PROM1*, *PGF* and *NCAM1* have not been reported to directly lead to PTC. We hypothesize that the interaction between them in the form of high-density subgraph may lead to PTC.

### 3.5. GO Term and KEGG Pathway Enrichment Analysis

The results of DEGs-related GO annotation in PTC are shown in Figure 5 and Figure 6. The GO can be divided into three functional groups: molecular function, which indicates molecular activities of gene products; biological process, whose pathways and larger processes made up of the activities of multiple gene products; and cellular component, where gene products are active.

We put the DEGs into the DAVID online tool. As is shown in Figure 5, in the biological procession, the up-regulated genes are mainly enriched in GO:0070062, GO:0005886, GO:0005615, and GO:0005576, which correspond to extracellular exosome, plasma membrane, extracellular space, and extracellular region, respectively. As is shown in Figure 6, the down-regulated genes are mainly enriched on GO:0005515, GO:0070062, GO:0005886, GO:0005615, and GO:0005576, which correspond to protein binding, extracellular exosome, plasma membrane, extracellular space, and extracellular region, respectively. Table 3 shows the number of genes enriched on each GO term, and this table only lists the genes from each GO term that are the same as the results calculated using the new measure NCM. From the results we can see that the top 40 genes identified using the NCM measure enrich in these terms described above.

By using the KOBAS online analysis database to analyze the pathway enrichment, we found that DEGs were mainly involved in 41 KEGG classical pathways, especially in cytokine-cytokine receptor interaction (12 genes), PI3K-Akt signaling pathway (12 genes) and metabolic pathways (24 genes). The three pathways with the largest number of genes connected to them are listed in Table 4. *p*-value < 0.05 is selected as the cut-off criterion to identify the enrichment pathway. These three pathways have been proven to participate in PTC. Zhang et al. [52] revealed DEGs of PTC are mainly involved in cytokine-cytokine receptor interaction pathway, which may regulate PTC growth and metastasis. Agrawal et al. [56] demonstrated that aberrant activation of the PI3K-Akt pathway plays a role in thyroid tumorigenesis. Metabolites such as neuroactive ligandreceptor interactions [57] in metabolic pathways vary widely between normal and diseased samples of PTC.

## 4. Conclusions

In conclusion, a research framework was proposed to identify the important characteristic genes which may be involved in PTC progression through the integrated analysis of multiple original microarray datasets. Based on the existing measures to quantify the topological properties of nodes, this paper proposed a new measure to comprehensively consider both local and global features of nodes. Experiments have demonstrated that this measure has important biological significance. Moreover, the availability of gene ontology and pathway enrichment analysis may provide evidence for cancer research. The results obtained in this paper may help to better understand the molecular mechanisms underlying PTC and provide a range of potential biomarkers. However, further experiments are needed to verify the results.

Our method has several advantages. First, the new measure we proposed is not simply to analyze one side of the local or global characteristics of each node, but to comprehensively analyze the topological properties of each node and the role it plays in the entire network. Second, the majority of existing studies focused on how a single characteristic gene contributes to the development of tumors in PTC, with limited research concerning the interaction of multi-genes. Herein, we analyzed the high-density subgraphs of the network constructed by DEGs to find interactions between multiple genes. Third, using network analysis methods makes the display of interactions between DEGs more intuitive and clear, and can significantly reduce the time of analysis and the difficulty of calculation.

Though our method is a beneficial exploration in detecting the DEGs, it still has several limitations, and needs further improvement and development. First, there are several parameters that need to be manually set, and no definitive standard to regulate the setting of logFC and *p*-value. Second, there are many studies to choose the DEGs at present, and no reasonable description to explain that the limma software is the best. These limitations motivate us to continue exploring in the future.

## Figures and Tables

**Figure 1 genes-10-00045-f001:**
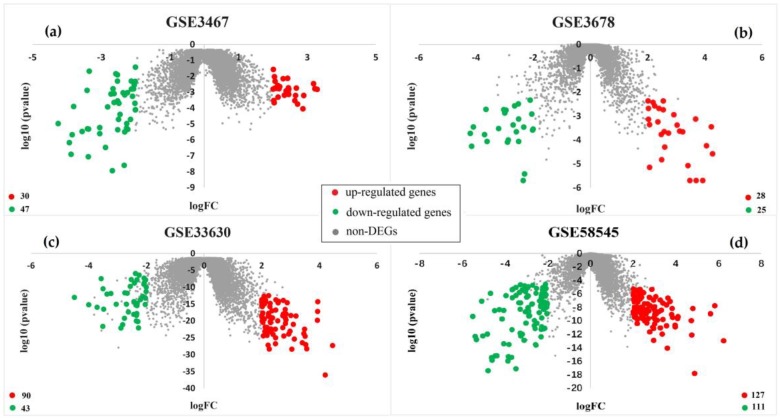
Volcano map showing differentially expressed genes (DEGs). (**a**–**d**) show DEGs of dataset GSE3467, GSE3678, GSE33630, and GSE58545, respectively.

**Figure 2 genes-10-00045-f002:**
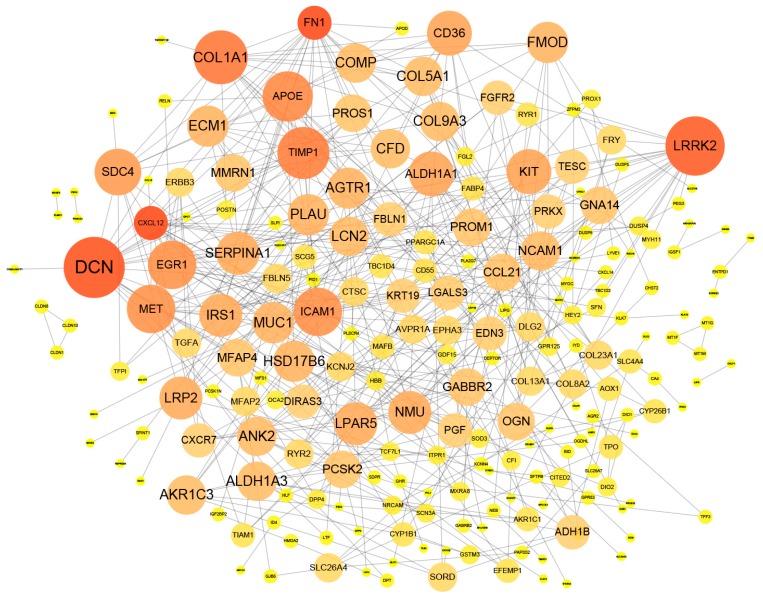
Interaction network of DEGs. The size and color of the node represent the importance of the node. The larger the node and the redder the color, the more neighbors are connected to the node.

**Figure 3 genes-10-00045-f003:**
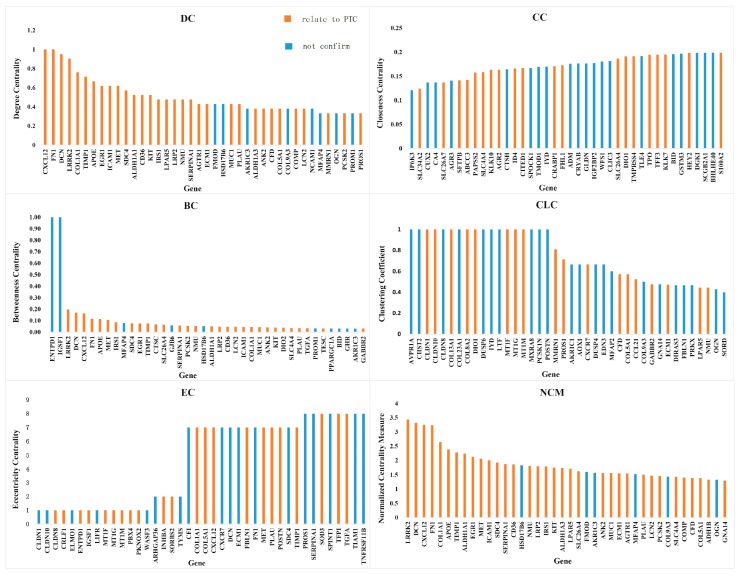
Histograms of topological property values of genes. The genes represented by the histograms of orange have been confirmed to be related to papillary thyroid carcinoma (PTC); the genes represented by the histograms of blue have not been confirmed to be related to PTC.

**Figure 4 genes-10-00045-f004:**
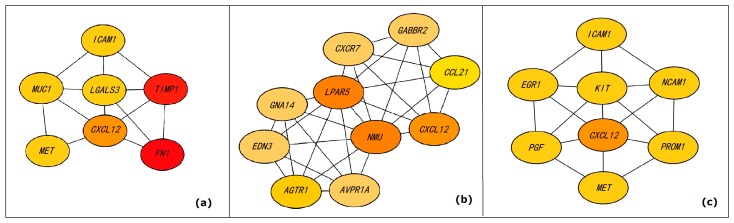
High-density subgraphs of DEGs interaction network. (**a**–**c**) show the high-density subgraphs

**Figure 5 genes-10-00045-f005:**
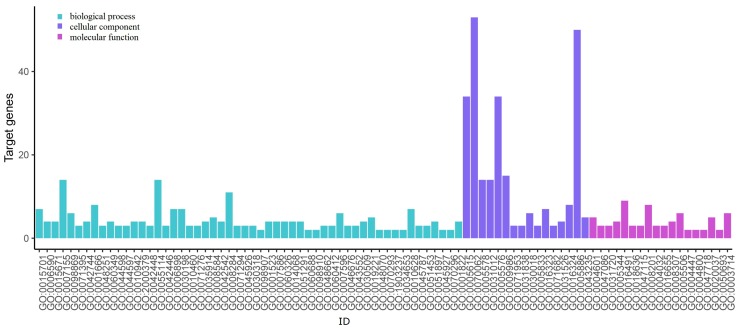
Up-regulated genes Gene Ontology (GO) enrichment analysis.

**Figure 6 genes-10-00045-f006:**
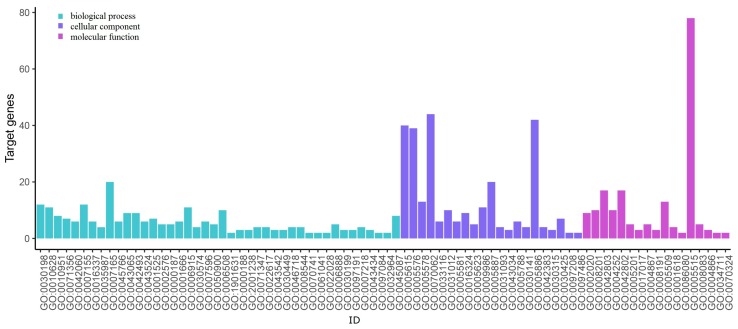
Down-regulated genes GO enrichment analysis.

**Table 1 genes-10-00045-t001:** Samples distribution of papillary thyroid carcinoma (PTC) dataset.

GEO	Platform	Normal	Tumor	Reference
GSE3467	GPL570	9	9	He et al. [9]
GSE3678	GPL570	7	7	He et al. [10]
GSE33630	GPL570	49	45	Tomás et al. [11]
GSE58545	GPL96	27	18	Rusinek et al. [12]

**Table 2 genes-10-00045-t002:** References of PTC-related genes in top 40 captured genes.

Measures	Count ^1^	Genes ^2^
Degree Centrality	32	*AGTR1* [34] *ALDH1A1* [30] *COL1A1* [21] *COMP* [25] *CXCL12* [26] *DCN* [32] *ECM1* [24] *FN1* [20] *ICAM1* [20] *IRS1* [26] *KIT* [24] *LCN2* [23] *LRP2* [33] *LRRK2* [2] *MET* [21] *MUC1* [29] *PLAU* [20] *SDC4* [2] *SERPINA1* [21] *TIMP1* [20] *CD36* [31] *EGR1* [38] *ALDH1A3* [37] *NMU* [36] *ANK2* [39] *APOE* [35] *CFD* [27] *COL5A1* [20] *LPAR5* [36] *PCSK2* [28] *PROS1* [2] *MMRN1* [27]
Closeness Centrality	20	*AGR2* [40] *CITED1* [2] *CRABP1* [41] *KLK10* [42] *KLK7* [25] *S100A2* [43] *SFTPB* [44] *SLC26A4* [30] *SLC34A2* [25] *TFF3* [28] *TMPRSS4* [45] *TPO* [41] *DIO1* [32] *ID4* [45] *PAPSS2* [39] *SLC4A4* [25] *ABCC3* [46] *FHL1* [38] *HEY2* [46] *SLC26A7* [47]
Betweenness Centrality	30	*ALDH1A1* [30] *COL1A1* [21] *CTSC* [48] *CXCL12* [26] *DCN* [32] *FN1* [20] *ICAM1* [20] *IRS1* [26] *KIT* [24] *LCN2* [23] *LRP2* [33] *LRRK2* [2] *MET* [22] *MUC1* [29] *PLAU* [21] *SDC4* [2] *SERPINA1* [22] *SLC26A4* [30] *TGFA* [41] *TIMP1* [19] *DIO2* [32] *CD36* [31] *ANK2* [39] *APOE* [35] *GABBR2* [24] *PCSK2* [28] *SLC4A4* [25] *TESC* [39] *EGR1* [38] *NMU* [36]
Clustering Coefficient	18	*CCL21* [49] *CLDN1* [20] *CLDN10* [50] *ECM1* [24] *DIO1* [32] *MT1G* [51] *CFD* [27] *COL13A1* [22] *COL5A1* [21] *CXCR7* [52] *GABBR2* [24] *MMRN1* [27] *PROS1* [2] *COL8A2* [53] *LPAR5* [36] *MT1F* [54] *MT1M* [54] *NMU* [36]
Eccentricity Centrality	22	*CLDN1* [20] *CLDN10* [50] *MT1G* [51] *COL1A1* [20] *CXCL12* [26] *DCN* [32] *ECM1* [24] *FN1* [20] *MET* [22] *PLAU* [20] *SDC4* [2] *TIMP1* [20] *SERPINA1* [22] *TGFA* [41] *TNFRSF11B* [22] *ELMO1* [55] *IGSF1* [55] *MT1F* [54] *SORBS2* [39,48] *COL5A1* [21] *CXCR7* [52] *PROS1* [2]
Normalized Centrality Measure	34	*ALDH1A1* [30] *CD36* [31] *COL1A1* [20] *COMP* [25] *CXCL12* [26] *DCN* [32] *ECM1* [24] *FN1* [20] *ICAM1* [20] *IRS1* [26] *KIT* [24] *LCN2* [23] *LRP2* [33] *LRRK2* [2] *MET* [22] *MUC1* [29] *PLAU* [21] *SDC4* [2] *SERPINA1* [22] *SLC26A4* [30] *TIMP1* [20] *AGTR1* [34] *ANK2* [39] *APOE* [35] *CFD* [27] *COL5A1* [21] *LPAR5* [36] *PCSK2* [28] *SLC4A4* [25] *ADH1B* [37] *GNA14* [37] *NMU* [36] *ALDH1A3* [37] *EGR1* [38]

^1^ Count: The number of genes that have been confirmed to be related to PTC from top 40 genes obtained by six measures. ^2^ Genes: Details of existing literature that described these genes to be involved in the production of PTC. DC: Degree Centrality; CC: Closeness Centrality; BC: Betweenness Centrality; CLC: Clustering Coefficient; EC: Eccentricity Centrality; NCM: Normalized Centrality Measure.

**Table 3 genes-10-00045-t003:** Gene Ontology (GO) enrichment analysis of DEGs.

DEGs	ID	Term	Count ^1^	Genes ^2^
Up-regulated	GO:0070062	extracellular exosome	53	*TIMP1 FN1 APOE COMP SERPINA1 NMU ECM1 COL5A1 LCN2 COL1A1 PLAU MET*
	GO:0005886	plasma membrane	50	*GNA14 AGTR1 SLC4A4 IRS1 CD36 KIT LRP2 SLC26A4 ANK2*
	GO:0005615	extracellular space	34	*FMOD KIT DCN CXCL12 OGN CFD CD36*
	GO:0005576	extracellular region	34	*FMOD DCN CXCL12 OGN COL9A3 CFD MFAP4*
Down-regulated	GO:0005515	protein binding	78	*APOE COL1A1 COMP ECM1 FN1 ICAM1 LRRK2 MUC1 NMU PCSK2 PLAU SDC4 SERPINA1 TIMP1 MET*
	GO:0070062	extracellular exosome	44	*SDC4 TIMP1 APOE COMP ALDH1A3 SERPINA1 FN1 MUC1 ICAM1 ECM1 COL5A1 LCN2 LRRK2 PLAU*
	GO:0005886	plasma membrane	42	*SDC4 APOE ICAM1 MET LPAR5 LRRK2 PLAU*
	GO:0005615	extracellular space	40	*TIMP1 PCSK2 APOE COMP SERPINA1 FN1 MUC1 ICAM1 ECM1 LCN2 COL1A1 LRRK2 PLAU*
	GO:0005576	extracellular region	39	*TIMP1 APOE COMP SERPINA1 NMU FN1 MET ECM1 COL5A1 LCN2 COL1A1 PLAU*

^1^ Count: the number of genes involved in the term. ^2^ Genes: genes in this term correspond to the top40 genes calculated using NCM.

**Table 4 genes-10-00045-t004:** Kyoto Encyclopedia of Genes and Genomes (KEGG) pathway analysis of DEGs associated with PTC.

Term	ID	Input Number	*p*-Value	Input Genes
Cytokine-cytokine receptor interaction	hsa04060	12	0.0002	*KIT MET CXCL12 LIFR CXCL14 ACKR3 TNFRSF11B TNFRSF12A CCL18 CCL21 GHR INHBA*
PI3K-Akt signaling pathway	hsa04151	12	0.0017	*MET FN1 RELN KIT FGFR2 LPAR5 COMP COL1A1 PGF IRS1 GHR COL9A3*
Metabolic pathways	hsa01100	24	0.0267	*AKR1C3 TYMS GALNT7 TPO ADH1B LIPG ALDH1A1 GALE OGDHL TUSC30 PLA2G73 SORD CSGALNACT1 PAPSS2 GATM CYP26B1 DGKI ALDH1A3 ALOX15B HGD TDO2 AOX1 HSD17B6 IMPA2*

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
