# Peer review of "Network Analyses of Integrated Differentially Expressed Genes in Papillary Thyroid Carcinoma to Identify Characteristic Genes"

_genes, 2019, doi:10.3390/genes10010045_

Round 1

Reviewer 1 Report

The manuscript entitled "Network Analyses of Integrated Differentially Expressed Genes in Papillary Thyroid Carcinoma to identify characteristic genes" proposes a novel network topology measure to identify genes involved in PTC. Despite being an interesting proposal, the study presents lack of methodological information and little in-depth comparison about existing measures and the proposed NCM measure. It is necessary to restructure the methodology and discuss the findings. Below are the comments and suggestions:

- It is suggested to the authors to add a table with BC, CC, DC and NCM measurements for a list of at least 40 differentially expressed genes (up and down) to reflect the importance of the new method for gene selection.

- If the proposed method is a new measure for the prioritization of disease-associated genes, a more complete description of how these genes were associated with diseases (which database used as a reference?) is needed since a more thorough search in PUBMED on certain genes described as "non-associated with PTC" could point to the opposite.

- It is not clear which were the main DEGs selected in the study and later used for network construction and extract the measurements. It would be interesting to see if the genes selected according to the best measures were also selected with higher FC. It is suggested to the authors to add a table with this information including p-value and fold-change. Also, in a Tumor vs Normal comparison it is expected to use an adjustment for the p-value (FDR, BH, Bonferroni) to increase the confidence of the selected genes.

- It is surprising how the authors achieved so few DEGs (5 genes) in the evaluation of the samples available in the GSE3467 study (9 tumor-normal paired samples) considering only p-value <0.05 and logFC > 1 and <-1. It is suggested to review this and other analysis.

- Figure 1 does not add much information since Table 2 already presents the numbers of DEGs. In addition, there is no indication of what the x-axis and y-axis represents.

- The authors should add in Figure 2 what the colors represent and what was the variation of connectivity (minimum and maximum) associated with node size.

- The results are poorly explored, both in the presentation and in the discussion of the data. A more critical scientific view regarding the importance of genes assigned as involved in PTC tumorigenesis is essential.

Author Response

Response to Reviewer 1 Comments

Point 1: It is suggested to the authors to add a table with BC, CC, DC and NCM measurements for a list of at least 40 differentially expressed genes (up and down) to reflect the importance of the new method for gene selection.

Response 1: Thank you for your comments. Based on your suggestion, we have list 40 differentially expressed genes, which are calculated by DC, BC, CC, CLC, EC and NCM measures, to reflect the importance of the NCM for gene selection. Since this table is too large, we show the same information in Figure 3. [Page 7 in manuscript]

Point 2: If the proposed method is a new measure for the prioritization of disease-associated genes, a more complete description of how these genes were associated with diseases (which database used as a reference?) is needed since a more thorough search in PUBMED on certain genes described as "non-associated with PTC" could point to the opposite.

Response 2: In the manuscript, we have described these disease-related genes which were detected using the NCM measure. These genes were searched in some traditional databases, such as HapMap and PUBMED. It is possible that some genes were proved to be related to PTC in recent years, but the classic databases may have not collected these genes. Therefore, we mainly proved by finding out if there existing papers documenting that the gene is related to PTC. [Page 7-8 in manuscript]

Point 3: It is not clear which were the main DEGs selected in the study and later used for network construction and extract the measurements. It would be interesting to see if the genes selected according to the best measures were also selected with higher FC. It is suggested to the authors to add a table with this information including p-value and fold-change. Also, in a Tumor vs Normal comparison it is expected to use an adjustment for the p-value (FDR, BH, Bonferroni) to increase the confidence of the selected genes.

Response 3: Thank you for your suggestions. We have described the selected DEGs in the manuscript: "A gene which satisfies conditions of P-value (adjusted by Benjamini & Hochberg) < 0.05 and logFC > 1 (FC, fold change) is identified as an up-regulated gene. A gene which satisfies conditions of P-value (adjusted by Benjamini & Hochberg) < 0.05 and logFC < -1 is identified as a down-regulated gene. All identified up-regulated genes and down-regulated genes are integrated into DEGs"; What's more, we used p-value and logFC to identify differentially expressed genes from four data sets respectively. Then, these genes are used to integrate together. The integrated data were used to consrtuct an interaction network and further detection. Instead of using p-value and logFC, we used NCM measure for detecting on the network which constructed by integrated data. Different from p-value and logFC, as a topological measure based on degree centrality, betweenness centrality and closeness centrality, NCM is used to analysis the topological property of network. P-value and logFC as well as NCM were used to analysis from two different perspectives; In addition, we used the Benjamini & Hochberg adjustment for the p-value to increase the confidence of the selected genes, but it is not described very clear in the manuscript, we have revised it. [Page 3 in manuscript]

Point 4: It is surprising how the authors achieved so few DEGs (5 genes) in the evaluation of the samples available in the GSE3467 study (9 tumor-normal paired samples) considering only p-value <0.05 and logFC > 1 and <-1. It is suggested to review this and other analysis.

Response 4: Samples in the GSE3467 data set are few. In the process of converting the probe name into the corresponding gene name, we found that multiple probe names corresponded to a same gene name. For this case, we took their median value. According to the calculation results, the occurrence of the same gene name is very common in GSE3467. Therefore, differentially expressed genes obtained are very few after taking the median values. [Page 5 in manuscript]

Point 5: Figure 1 does not add much information since Table 2 already presents the numbers of DEGs. In addition, there is no indication of what the x-axis and y-axis represents.

Response 5: Thank you for your careful negotiation. We have been redrawn Figure 1 and deleted the table in the new manuscript. [Page 5 in manuscript]

Point 6: The authors should add in Figure 2 what the colors represent and what was the variation of connectivity (minimum and maximum) associated with node size.

Response 6: We have explained in the revised manuscript: "The size and color of the node represent the importance of the node. The larger the node and the redder the color, the more neighbors are connected to the node". [Page 6 in manuscript]

Point 7: The results are poorly explored, both in the presentation and in the discussion of the data. A more critical scientific view regarding the importance of genes assigned as involved in PTC tumorigenesis is essential.

Response 7: In the manuscript, we have described in detail the importance of the genes which were detected from the network using NCM. [Page 7-8 in manuscript]

Reviewer 2 Report

In this work, Junliang Shang et al present a new metrics called Normalized Centrality Measurement (NCM) to identify characteristic genes from differentially expressed gene networks in papillary thyroid carcinoma (PTC). This measurement combined both local and global topological information, which identified more critical genes related to PTC. It is an interesting idea to integrate traditional measures into a more comprehensive metrics. I do have a few comments on the experimental design and validation.

1. In table1, it is strange to see GSE3678 have such a high number of DEGs. Usually, the more samples you have, the stronger the statistical power you have to detect significant differences. Given only 7 patients in GSE3678, it is surprising to see a nearly 50-fold increase in the DEG numbers comparing to GSE3476, which has 9 patients. How are these sample collected and handled?

2. Independent validation is needed. Can you observe similar pattern if you use TCGA thyroid expression data?

3. In figure 3, why select top 20 as cutoff? It is not fair for simply counted the number of related genes in top 20 to claim NCM is better than the others. An enrichment test should be used to support this point.

4. In figure 3 EC panel, it seems a lot of genes share equal score. I suspect more genes share a score of 2 as well.

5. In line 290, we put the 265 differentially expressed genes into.. Where does 265 come from? In line 197, it is 345 DEGs.

Minor points:

6. HC-HDSD appears first time at line 71 without explanation of full name.

7. Line 214, keep font consistent.

Author Response

Response to Reviewer 2 Comments

Point 1: In table1, it is strange to see GSE3678 have such a high number of DEGs. Usually, the more samples you have, the stronger the statistical power you have to detect significant differences. Given only 7 patients in GSE3678, it is surprising to see a nearly 50-fold increase in the DEG numbers comparing to GSE3476, which has 9 patients. How are these sample collected and handled?

Response 1: Thank you for your comments. These four data sets are downloaded from the GEO database. We converted the probe names to corresponding gene symbols, performed log2 transformation and z-score normalization. There are multiple probe names corresponded to a same gene name in the process of converting the probe name. For this case, we took their median values. GSE3467 contains more of the same gene names than GSE3678, and differentially expressed genes obtained are fewer than GSE3678 after taking the median values. Therefore, differentially expressed genes obtained are very different between these two data sets. [Page 5 in manuscript]

Point 2: Independent validation is needed. Can you observe similar pattern if you use TCGA thyroid expression data? 

Response 2: Thank you for your careful negotiation. In our current study, we mainly analyze the data downloaded from the GEO database. This manuscript is concentrated on proposing a research framework to integrate and mine multiple data sets from the GEO database. We have not yet covered the data in TCGA. Your suggestion is very helpful. In the next study, we will download and do more in-depth research on the data sets from TCGA. If you are interested, you can look forward to our next step.

Point 3: In figure 3, why select top 20 as cutoff? It is not fair for simply counted the number of related genes in top 20 to claim NCM is better than the others. An enrichment test should be used to support this point.

Response 3: According to the comments of the first reviewing expert, top 20 is few, and we adjusted to top 40 in the revised manuscript. 13, 11, 9, 5, 5, and 5 PTC-related genes were obtained among the top 20 genes by using NCM, DC, BC, CC, CLC and EC measures. The accuracy is 65%, 55%, 45%, 25%, 25%, 25%, respectively. And 28, 27, 20, 12, 18, and 17 PTC-related genes were obtained among the top 40 genes by using NCM, DC, BC, CC, CLC and EC measures. The accuracy is 70%, 67.5%, 50%, 30%, 45%, 42.5%, respectively. So, we chose top 40 as cutoff. What's more, we have made corresponding adjustments to the results of the GO enrichment analysis. [Page 6-8, 10 in manuscript]

Point 4:In figure 3 EC panel, it seems a lot of genes share equal score. I suspect more genes share a score of 2 as well.

Response 4: In the revised manuscript, we selected the top 40 genes, and redrawn Figure 3, In Figure 3 EC panel, it indeed seems a lot of genes share a score of 2, but not all are 2. [Page 7 in manuscript]

Point 5: In line 290, we put the 265 differentially expressed genes into..Where does 265 come from? In line 197, it is 345 DEGs.

Response 5: We have been revised in the new manuscript: "A complex interaction network is obtained from STRING database to describe interactions between genes, with a total of 345 DEGs including 173 up-regulated genes and 172 down-regulated genes. The nodes in the network represent genes. After removing the duplicate values and the individual nodes that do not connect to other nodes, there is a total of 265 nodes in this network." [Page 6 in manuscript]

Point 6: HC-HDSD appears first time at line 71 without explanation of full name.

Response 6: Thank you for your comments and we have been revised in the new manuscript: [Page 2 in manuscript]

Point 7: Line 214, keep font consistent.

Response 7: Thank you for your careful negotiation and we have been revised in the new manuscript. [Page 5 in manuscript]

Round 2

Reviewer 1 Report

1. The authors reported in response 3 "Different from p-value and log CF, as a topological measure based on degree centrality, betweenness centrality and closeness centrality, NCM is used to analyze the topological property of network. were used to analysis from two different perspectives". Considering that the difference between what represents logFC and p-value compared to network topology measures is clear, the use of such measures is influenced by the genes selected and these genes are the reference for the multiple comparison among network topology measures. If the genes were previously selected with adjusted p-value (BH) a table with the top differentially expressed genes should contain this information. If the inclusion of this material exceeds the number of words and / or table according to the guideline of this journal, these data should be sent separately in response to what was previously mentioned.

2. It is quite acceptable the justification that the genes were manually searched to prove the relationship with PTC but this study design is questionable to explore the relationship between hundreds of differentially expressed genes and PTC in a study that proposes to compare measures of networks with the aim of presenting a more efficient method. Authors are encouraged to use well-annotated and recently updated databases to construct a reference list that can be benchmarked for multiple comparisons. The search for these genes in scientific articles is a good strategy for capturing recently published genes but this should not be the only method to associate genes and diseases. Further, authors reported that HapMap, a catalog of common genetic variants (SNPs), was used but no further consideration as to why this database was used, among many others, was presented in the article.

3. Even considering the small amount of available samples (9 tumor-normal paired samples) and multiple probe names corresponding to the same gene name, an analysis made for the GSE3467 study with the GEO2R scrip available from NCBI (considering Biobase, GEOquery and limma libraries ) shows that more than 1000 genes (adjusted-pvalue by BH and logFC> 1 and <-1) were selected (~ 800 unique gene symbols) (attached file). Even considering that the GEO2R does not provide a paired analysis between Tumor / Normal and that each researcher processes, normalizes and analyzes data according to the criteria that considered most relevant and technically acceptable this difference observed between what the GEO2R makes available and the number of genes presented in the manuscript (5 genes) suggest a review of the analysis presented or an explanation to justify such a difference. The selection of more DEGs could suggest more genes associated with PTC with a prominence in the topological structure of networks, increasing the potential of the results obtained in this study.

Author Response

Response to Reviewer 1 Comments

Point 1: The authors reported in response 3 "Different from p-value and log CF, as a topological measure based on degree centrality, betweenness centrality and closeness centrality, NCM is used to analyze the topological property of network. were used to analysis from two different perspectives". Considering that the difference between what represents logFC and p-value compared to network topology measures is clear, the use of such measures is influenced by the genes selected and these genes are the reference for the multiple comparison among network topology measures. If the genes were previously selected with adjusted p-value (BH) a table with the top differentially expressed genes should contain this information. If the inclusion of this material exceeds the number of words and / or table according to the guideline of this journal, these data should be sent separately in response to what was previously mentioned.

Response 1: We are thankful for this strict comment. The following table contains adjusted p-value (BH) and P-value of the genes that selected by NCM measure. As mentioned before, instead of using P-value and logFC, we used NCM measure for detecting on the network which constructed by integrated data. Different from P-value and logFC, as a topological measure based on degree centrality, betweenness centrality and closeness centrality, NCM is used to analysis the topological property of network. P-value and logFC as well as NCM were used to analysis from two different perspectives. Therefore, the genes that selected by calculating the NCM measure belong to the DEGs (P-value < 0.05 and |logFC| > 2), but do not necessarily have the value of the minimum P-value or the maximum |logFC|. Figure 1 shows the correlation of pvalue and NCM, figure 2 shows the correlation of logFC and NCM. As shown in the figures, we calculate the Pearson correlation coefficient and the Spearman correlation coefficient, respectively. It can be seen that the relationship between them is very weak.

Table s. P-value, logFC and NCM values of top 40 DEGs.

Genes

adj.P.Val

P.Value

logFC

NCM

LRRK2

9.4600E-13

3.0600E-14

-3.3637

3.4367

DCN

5.0450E-04

4.9550E-05

2.2100

3.3221

CXCL12

1.3200E-04

9.8300E-06

2.8200

3.2511

FN1

1.0365E-16

5.5750E-20

-3.8550

3.2336

COL1A1

1.4900E-06

4.6100E-08

-3.3200

2.6382

APOE

7.8200E-04

8.2200E-05

-2.2400

2.3872

TIMP1

8.4400E-23

1.7900E-25

-2.2709

2.2849

ALDH1A1

1.9000E-13

5.3200E-15

2.0849

2.2409

EGR1

8.1500E-05

5.5100E-06

2.0500

2.1305

MET

1.2200E-13

1.8100E-16

-3.1000

2.0654

ICAM1

4.1100E-08

6.9400E-10

-2.0200

2.0101

SDC4

1.2700E-11

5.0900E-14

-2.2700

1.9313

SERPINA1

3.0720E-16

2.3700E-19

-4.6200

1.8799

CD36

3.8125E-09

4.6323E-11

3.0050

1.8624

HSD17B6

5.6850E-09

4.4150E-10

2.1660

1.8352

NMU

4.4600E-04

4.1900E-05

-2.9300

1.8109

LRP2

1.0700E-07

2.0500E-09

2.0800

1.7997

IRS1

1.3300E-09

1.1900E-11

2.3700

1.7957

KIT

9.8200E-22

3.0200E-24

3.1002

1.7503

ALDH1A3

8.1200E-22

2.3600E-24

-2.8192

1.7384

LPAR5

7.8800E-05

2.9605E-07

-2.0019

1.7084

SLC26A4

3.2600E-09

2.3400E-10

2.4597

1.6251

FMOD

3.2400E-10

2.2100E-12

2.1400

1.6002

AKR1C3

2.9200E-07

6.7300E-09

2.4000

1.5700

ANK2

1.2300E-08

1.7200E-10

3.0100

1.5643

MUC1

2.0355E-06

7.0550E-08

-3.0400

1.5610

ECM1

3.2000E-17

3.6700E-19

-3.4804

1.5524

AGTR1

1.8400E-05

1.0650E-06

2.1405

1.5505

MFAP4

3.9900E-08

6.6300E-10

3.4000

1.5249

PLAU

4.4950E-08

7.8000E-10

-3.3400

1.5048

LCN2

2.2200E-05

1.1700E-06

-3.2700

1.4678

PCSK2

5.1419E-04

5.6678E-05

-2.9150

1.4584

COL9A3

2.5900E-15

4.5400E-17

2.8892

1.4365

SLC4A4

5.8400E-24

8.7500E-27

2.3221

1.4267

COMP

5.4700E-15

1.0500E-16

-2.4292

1.4023

CFD

5.3900E-25

5.4800E-28

2.1880

1.3859

COL5A1

6.0300E-05

3.8100E-06

-2.0600

1.3853

ADH1B

9.5100E-28

2.9600E-31

3.5313

1.3343

OGN

1.7313E-15

3.1130E-17

2.2969

1.3271

GNA14

1.0600E-09

9.0200E-12

2.6100

1.2960

Fig 1. The correlation of P-value and NCM.

Fig 2. The correlation of logFC and NCM.

Point 2: It is quite acceptable the justification that the genes were manually searched to prove the relationship with PTC but this study design is questionable to explore the relationship between hundreds of differentially expressed genes and PTC in a study that proposes to compare measures of networks with the aim of presenting a more efficient method. Authors are encouraged to use well-annotated and recently updated databases to construct a reference list that can be benchmarked for multiple comparisons. The search for these genes in scientific articles is a good strategy for capturing recently published genes but this should not be the only method to associate genes and diseases. Further, authors reported that HapMap, a catalog of common genetic variants (SNPs), was used but no further consideration as to why this database was used, among many others, was presented in the article.

Response 2: We are thankful for this strict comment. We first compare the selected genes with the PTC genetic data contained in the GeneCard database. In Table 2, genes with bold font have been recorded in GeneCard database. As mentioned before, it is possible that some genes were proved to be related to PTC in recent years, but the GeneCard databases may have not collected these genes. Therefore, we mainly proved by finding out if there existing papers documenting that the gene is related to PTC. In the process of searching literature which are recorded that genes related to PTC, we found that there are records about PTC caused by SNP, such as rs2577301, rs1800215 and so on. We performed a simple search in the NCBI and HapMap databases and found that rs2577301 is on the FN1 gene as well as rs1800215 is on the COL1A1 gene. FN1 and COL1A1 gene are also contain in our selected genes. Since this manuscript is concentrated on constructing and analysis the network at the gene level, there is no more in-depth analysis of related SNPs. [Page 7-8 in manuscript]

Point 3: Even considering the small amount of available samples (9 tumor-normal paired samples) and multiple probe names corresponding to the same gene name, an analysis made for the GSE3467 study with the GEO2R scrip available from NCBI (considering Biobase, GEOquery and limma libraries ) shows that more than 1000 genes (adjusted-pvalue by BH and logFC > 1 and <-1) were selected (~ 800 unique gene symbols) (attached file). Even considering that the GEO2R does not provide a paired analysis between Tumor / Normal and that each researcher processes, normalizes and analyzes data according to the criteria that considered most relevant and technically acceptable this difference observed between what the GEO2R makes available and the number of genes presented in the manuscript (5 genes) suggest a review of the analysis presented or an explanation to justify such a difference. The selection of more DEGs could suggest more genes associated with PTC with a prominence in the topological structure of networks, increasing the potential of the results obtained in this study.

Response 3: We are thankful for this strict comment. The pre-processing methods we used were not appropriate and resulted in biased results. In the manuscript, we removed the pre-processing method and referred to the results in GEO2R to adjust the experimental results. At the same time, in order to facilitate the processing of the subsequent process and make it more statistically significant, we adjust the threshold to adjusted P-value by BH and logFC > 2 and <-2. In the reversed manuscript, 77, 53, 123, 238 differentially expressed genes are obtained by the limma package from the above four datasets, among which there are 30, 28, 90, and 127 up-regulated genes as well as 47, 25, 43, and 111 down-regulated genes, respectively, respectively. [Page 5 in manuscript]

Reviewer 2 Report

In comment 3, changing cutoff from 20 to 40 does not solve the problem. Perform enrichment test is an fair way to make the comparison (e.g. use algorithm similar to GSEA). 

In comment 4, if they have the same score, how did you select which ranks in top20? In this version, what about the score of the 41st gene? I suppose the score is the same as 40th gene, and you simply select it by alphabetical order. This is not correct. Please refer to comment 1 to solve this problem.

Author Response

Response to Reviewer 2 Comments

Point 1: In comment 3, changing cutoff from 20 to 40 does not solve the problem. Perform enrichment test is an fair way to make the comparison (e.g. use algorithm similar to GSEA).

Response 1: We are thankful for this comment. In the revised manuscript, we performed GO and KEGG enrichment analysis of DEGs with PTC, which is similar to GSEA algorithm. [Page 11 in manuscript]

Point 2: In comment 4, if they have the same score, how did you select which ranks in top20? In this version, what about the score of the 41st gene? I suppose the score is the same as 40th gene, and you simply select it by alphabetical order. This is not correct. Please refer to comment 1 to solve this problem.

Response 2: We are thankful for this comment. What we want to compare is that the ratio of the PTC-related genes among the top 40 genes selected by each measure is larger, and we think that the more biologically significant this measure is. There are 34, 31, 30, 20, 18, and 22 genes have been confirmed in the existing literature to be related to the production of PTC by calculating the NCM, DC, BC, CC, CLC and EC measures of topological properties. The accuracies are 85%, 77.5%, 75%, 50%, 45%, 55%, respectively. Considering that the 41st and 40st genes of EC have the same value, and this gene is not recorded in GeneCard, the accuracy of these 41 genes is about 53%. At the same time, we found that the EC values of the 33st to 97st genes were all 8. We found in the GeneCard database that the accuracy of these 97 genes is about 51%, which is smaller than the proportion of the top 40.

Round 3

Reviewer 1 Report

Minor revision

- Authors should include the version of the software that was used online, e.g. KOBAS (2.0, 3.0?).

- A thorough review should be made in the article so that some errors can be corrected, e.g. "Table 1. Samlpes distribution of papillary ..."

Reviewer 2 Report

Double check for grammar and formatting like extra spaces and spelling errors.